# Relationship between Testosterone and Sarcopenia in Older-Adult Men: A Narrative Review

**DOI:** 10.3390/jcm11206202

**Published:** 2022-10-20

**Authors:** Kazuyoshi Shigehara, Yuki Kato, Kouji Izumi, Atsushi Mizokami

**Affiliations:** Department of Integrative Cancer Therapy and Urology, Kanazawa University Graduate School of Medical Science, 13-1, Takaramachi, Kanazawa 920-8641, Japan

**Keywords:** testosterone, sarcopenia, frailty, muscle

## Abstract

Age-related decline in testosterone is known to be associated with various clinical symptoms among older men and it is possible that the accompanying decline in muscle mass and strength might lead to a decline in motor and physical functions. Sarcopenia is an important pathophysiological factor associated with frailty in older adults and is diagnosed in older adults as a decrease in muscle strength, muscle mass, and walking speed, which can lead to a significant decline in the quality of life and shortened healthy life expectancy. Testosterone directly interacts with the androgen receptor expressed in myonuclei and satellite cells and is also indirectly associated with muscle metabolism through various cytokines and molecules. Currently, significant correlations between testosterone and frailty in men have been confirmed by numerous cross-sectional studies. Many randomized control studies have also supported the beneficial effect of testosterone replacement therapy (TRT) on muscle volume and strength among men with low to normal testosterone levels. In the world’s aging society, TRT can be a tool for preventing the onset of sarcopenia in older-adult men. This narrative review aims to show the relationship between the decline in testosterone with age, sarcopenia, and frailty, as well as the effects of testosterone replacement therapy on muscle mass and strength.

## 1. Introduction

The aging of the population has been become a worldwide problem and maintaining and even improving the quality of life (QOL) of middle-aged and older-adult men has become an important issue. As the population ages, healthy life expectancy becomes increasingly more important than mean life expectancy. Life expectancy refers to the period from birth to death and includes periods requiring long-term care, while healthy life expectancy indicates the period without significant health issues in daily life. In many developed countries, the difference between the mean life expectancy and the healthy life expectancy tends to be greater and extending healthy life expectancy has become a clinical concern.

In men, serum testosterone levels decrease with age by 2–3% annually, a decline associated with specific symptoms of late-onset hypogonadism (LOH) syndrome [1], whose various clinical signs and symptoms include decreased libido and sexual desire, muscle weakness, increased visceral fat, obesity, osteoporosis, deterioration of insulin resistance, and dyslipidemia, which are significantly associated with aging [1,2,3,4]. These clinical signs and symptoms can often impair the QOL of middle-aged and older-adult men and are becoming a serious issue in the present aging society. Testosterone replacement therapy (TRT) is a widely accepted tool for improving these clinical conditions in hypogonadal men and its clinical use has increased substantially over the past several years [1].

Sarcopenia is an important pathophysiological factor of frailty in older adults and is diagnosed with a decrease in muscle strength, muscle mass, and low physical performance [5]. Sarcopenia and frailty are significantly associated with an increased risk of falls and fractures in older-adult men, which can lead to a serious decrease in QOL and shortening of healthy life expectancy [6,7]. Preventive measures are, therefore, required.

Testosterone is an important hormone for maintaining skeletal muscle mass and strength in men and numerous previous studies have suggested that testosterone deficiency is significantly associated with the onset of sarcopenia. The present review summarizes the current evidence on the relationship between testosterone and sarcopenia/frailty. The review also investigated whether TRT for hypogonadal men with sarcopenia can improve muscle mass, strength, and physical function.

## 2. Materials and Methods

A review of the PubMed, MEDLINE, and EMBASE databases was conducted to search for original articles, systematic reviews, and meta-analyses under following keywords: “testosterone” or “hypogonadism” or “sarcopenia” or “frailty” or “muscle”. There were no limitations in terms of language, publication status, or study design. Papers published between January 1990 and October 2020 were collected. We also checked the references of systematic reviews and meta-analyses carefully to identify additional original articles for inclusion. Two reviewers screened the search results and the data were collected in June 2022.

The papers suitable for the topic “testosterone and sarcopenia” from the journal databases were chosen for the present analysis. For “efficacy of testosterone replacement therapy in sarcopenia”, we reviewed papers published since 2010.

## 3. Sarcopenia: Definition and Etiology

One of the phenomena of human aging is the progressive decline in skeletal muscle mass, which can result in negative effects on physical fitness and function. The prevalence of sarcopenia is ~5–50% in older adults aged 65 years and older [8,9,10]. In 1989, Irwin Rosenberg proposed the term “sarcopenia” (from the Greek “sarx” for flesh and “penia” for deficiency) for age-related loss of muscle mass [11]. Currently, sarcopenia is defined as age-related loss of skeletal muscle mass and strength. Various differing definitions of sarcopenia have been proposed; however, there is still no widely accepted definition [12]. Baumgartner et al. first defined sarcopenia as a decline of less than 2 standard deviations below the mean of a young reference group in appendicular skeletal muscle mass [13]. Since then, the algorithm proposed by the European Working Group on Sarcopenia in Older People (the presence of both low muscle mass and low muscle strength or performance) [14] and by the Foundation for the National Institutes of Health sarcopenia project (appendicular lean mass adjusting for body mass index to define low muscle mass) [15] have been generally used for diagnosing sarcopenia. Recently, the Asian working group for sarcopenia suggested an alternative algorithm (Figure 1) [16]. This Asian consensus has plenty in common with the European consensus and the cases with decreased walking speed or grip strength are defined as individuals with decreased muscle performance. Those cases accompanied by decreased muscle mass are defined as having sarcopenia. According to the Asian consensus, the reported prevalence for sarcopenia is 16.5% for men and 19.9% for women in Japan [17], which is likely to be approximately similar to that reported previously in other countries [18,19,20].

For certain cases, the cause of the sarcopenia can be clearly identified, whereas no clear cause can be determined in other cases. Sarcopenia caused only by aging is classified as “primary” (age related) and sarcopenia caused by activities of daily living, nutrition, and illness is classified as “secondary sarcopenia” (Table 1). The most common cause of muscle weakness in older adults is age-related muscle atrophy. In general, skeletal muscle decreases by 25–30% and muscle strength decreases by 30–40% in individuals in their 70s when compared with those in their 20s, with muscle mass decreasing by approximately 1–2% every year after 50 years of age [21]. In addition, older-adult men have an increased risk of developing sarcopenia through various factors, such as lifestyle changes, less exercise, more physical illnesses (severe organ failure, neuromuscular disease, inflammatory disease, malignant tumors, etc.), undernourishment, and appetite loss. Therefore, the etiology of sarcopenia is often assumed to be multifactorial [22]. Sarcopenia develops concurrently with changes in hormones (testosterone and growth hormones) and inflammatory cytokines involved in the muscle metabolism due to these causes. In particular, testosterone is significantly correlated with maintaining bone strength, muscle mass, and muscle strength among men and it has been found that the pathogenesis of sarcopenia in men might be associated with testosterone decline with aging.

## 4. Testosterone and Sarcopenia

### 4.1. Testosterone and Muscle Metabolism

More than 95% of serum testosterone is produced by the Leydig cells of the testes through stimulation by LH from the pituitary gland in males [23]. Testosterone is mostly bound to sex-hormone-binding globulin or albumin and 1–2% exists in free form; however, it binds loosely to albumin and can easily become free form [24]. Free testosterone (FT) is taken up into target cells through the cell membrane and binds to androgen receptor (AR) in the cytoplasm. Testosterone bound to AR is converted to dihydrotestosterone by 5α-reductase. Testosterone and DHT bind to the same AR to form a dimeric complex and this dimer binds to specific sites on DNA and activates target genes, resulting in the expression of androgenic actions [25]. In general, the length of the CAG repeats present in the AR gene shows an inverse relationship, with AR susceptibility and length of their repeats differing between the races, with the length increasing in the order of black people, white people, and Asian people [26].

There are numerous ARs in muscle tissue and testosterone plays an important role in maintaining muscle mass and strength. It is, therefore, logical to assume that age-related testosterone decreases are closely associated with the onset of sarcopenia in men. Conversely, the anabolic effects of testosterone on muscle hypertrophy have been well established [27].

In human studies, testosterone directly interacts with AR expressed in myonuclei and satellite cells [28,29], which is a major source for the establishment of hypertrophying muscle fibers (Figure 2). Testosterone has a potential effect on myogenesis and muscle hypertrophy by increasing protein synthesis and inhibiting protein degradation in muscle cells [30,31] and then promoting mitotic activity and differentiation of satellite cells [29,32]. Numerous in vitro studies have demonstrated the anabolic actions of testosterone through increases in insulin-like growth factor-1 expression [33,34], beta-catenin and T-cell factor-4 pathway signaling [35], regulation of peroxisome proliferator-activated receptor-gamma coactivator 1 alpha, and p38 mitogen-activated protein kinases [36] and stimulating the hypertrophy of L6 myoblasts in a signal cascade dependent on Ark and mammalian target of rapamycin [37]. The mechanism by which low testosterone levels cause muscle atrophy is also being clarified. The catabolic action of testosterone has been described through the enhancement of muscle atrophy-F-box (atrogin-1) and muscle RING-finger protein-1 expression [38]. Moreover, an increase in hypertrophied visceral fat due to testosterone decline contributes to an increase in certain inflammatory cytokines, such as interleukin-6 and tumor necrosis factor (TNF-α), which have catabolic effects on skeletal muscle [39].

### 4.2. Clinical Effects of Testosterone for Muscle

Androgen deprivation therapy (ADT) for prostate cancer can result in decreased muscle mass and muscle weakness. A prospective study that included 79 patients with prostate cancer and employed a 12-month ADT reduced the participants’ lean body mass by 3.8% and increased their body fat percentage by 11% [40]. According to a report examining 39 patients with prostate cancer, the muscle mass of the rectus femoris, sartorius, and quadriceps estimated using computed tomography after ADT for 14–20 weeks was 21.8% and 15.4%, and 16.6%, respectively [41]. In general, patients with prostate cancer who undergo ADT are reported to have 3.0–6.0% lower muscle mass and 15–17% lower muscle strength than healthy individuals of the same age [42]. These findings suggest that testosterone decline with age is a trigger for muscle loss among older-adult men.

Several studies have demonstrated an association between serum testosterone levels and muscle mass and strength in men [43,44,45,46,47,48,49]. Appendicular muscle mass was significantly correlated with serum FT levels in non-Hispanic white men in a cross-sectional study from the New Mexico Aging Process Study [43]. In 403 men from The Netherlands, bioavailable testosterone and luteinizing hormone had a significant correlation with grip and leg extensor strength [44]. The MINOS cohort study that included 845 French men also found that the group with the lowest appendicular muscle mass had a significantly lower FT level [45]. A previous National Health and Nutrition Examination Survey study of men revealed that higher testosterone levels at physiologic levels were associated with higher body lean mass and lower body fat mass [46]. A cross-sectional study that included 922 men aged over 60 years found that weaker muscle strength was observed in the men within the lowest tertile of FT compared with those in the highest tertile (adjusted odds ratio: 2.28; 95% CI 1.33–3.91) [47]. A recent study that investigated the association between serum testosterone levels and body composition among 3875 men in China found a positive correlation between testosterone levels and appendicular lean mass index [48]. A recent systematic review also reported that testosterone could have a potential effect on muscle mass and strength [49]. Although there are a few studies that failed to identify an effect for testosterone on muscle strength [50,51], current evidence has likely established a positive correlation between testosterone and muscle condition.

### 4.3. Evidence of Testosterone Decrease in Frailty/Sarcopenia

Numerous cross-sectional studies have confirmed significant correlations between testosterone and frailty in men [52,53,54]. A Massachusetts cohort study that included 646 men aged 50–86 years investigated the relationship between testosterone and frailty and its components [52]. Although no association was observed between total testosterone (TT) or FT levels and the frailty phenotype, there was a significant association between TT levels and the frailty components of grip strength and physical activity. Cross-sectional data from the Toledo Study for Healthy Aging that included 552 men showed that the risk of frailty decreases linearly with testosterone levels (adjusted OR 2.9 (95% CI 1.6–5.1) and 1.6 (95% CI 1.0–2.5) in TT and FT, respectively) [53]. Another cross-sectional study based on data from the Longitudinal Aging Study Amsterdam (LASA) that included 623 men also suggested a potential correlation between low TT or bioavailable testosterone levels and impaired mobility and low muscle strength in men [54]. In a study of 461 individuals aged 60 years and older, a low FT value (<243 pmol/L) was a significant risk factor for developing frailty [55]. A recent meta-analysis that included 11 studies reported that TT (OR 1.37, 95% CI 1.09–1.72) and FT (OR 1.55, 1.06–2.25) were significantly associated with frailty in older men [56].

A number of longitudinal studies have found an equivocal future risk for developing frailty and sarcopenia due to low baseline testosterone [57,58,59]. A longitudinal study that included 957 community-dwelling adult men in Japan demonstrated that low calculated FT (OR 2.14, 95% CI 1.06–4.33) and FT (OR 1.83, 95% CI 1.04–3.22) were associated with the onset of sarcopenia [57]. Another report that included 1445 men from the Framingham Offspring Study revealed that low FT levels were significantly associated with the incidence of mobility, limiting its progression, but was not associated with subjective health, usual walking speed, or handgrip strength after 6.6 years of follow-up [58]. A longitudinal study that included 486 men from LASA and 1071 well-functioning men from the Health, Aging and Body Composition study demonstrated that baseline FT was not associated with changes in physical performance, walking speed, or muscle strength after 3 years of follow-up [59]. Further studies are needed to conclude whether low testosterone levels predict the progression and development of incident frailty and sarcopenia.

## 5. Efficacy of Testosterone Replacement Therapy for Sarcopenia

### 5.1. Indication of Testosterone Replacement Therapy

The indication of TRT requires the presence of low serum testosterone level. However, the cut-off value of serum testosterone for a diagnosis of hypogonadism is still controversial, with multiple international societies’ recommendations [60]. The diagnosis of hypogonadism for the recommendation of TRT in the guidelines of the Consensus Committee of the American Urological Association (AUA) is TT ≤ 3.0 ng/mL [61]. On the other hand, according to the International Society for Sexual Medicine (ISSM), the International Society for the Study of Aging Males (ISSAM), and the European Association of Urology (EAU), serum TT levels above 12 nmol/L (346 ng/dL) are normal and TT levels below 8 nmol/L (231 ng/dL) indicate hypogonadism, meaning that TRT may be appropriate [3,4,62,63,64]. In cases with borderline TT values of 8–12 nmol/L, hypogonadism should be diagnosed with calculated FT values.

### 5.2. Efficacy of Testosterone Replacement Therapy for Sarcopenia

The randomized controlled trials (RCTs) published since 2010 are summarized in Table 2 [50,65,66,67,68,69,70,71,72,73,74,75,76,77,78,79,80,81,82]. Their results varied by target population, type of testosterone formulation, and testosterone dosage. Many of the RCTs investigated the effects on muscle of TRT among men with low to normal testosterone levels and 15 of 19 RCTs supported certain merits of TRT on muscle volume [50,67,68,69,70,72,73,74,75,76,77,79,80,81,82]. The other four studies, however, failed to demonstrate that TRT contributes to improving muscle mass or strength [65,66,71,78]. In one of the RCTs, the study population consists of patients with opioid-induced hypogonadism, which was a specific population differing from the LOH syndrome [65]. The other three studies investigated the efficacy of exercise and/or diet added to TRT on muscle mass and strength but did not study the direct effects of TRT in isolation [66,72,78]. These findings suggest that monolithic TRT for hypogonadal men can contribute to improving muscle mass and strength. However, certain clinical interventions, such as exercise and diet added to TRT, are likely to be the most important factor for maintaining muscle function among older-adult men.

A recent meta-analysis demonstrated that TRT produced an increase in lean body mass of 2.54 kg (95% CI 1.27–3.80; *p* < 0.001) and an increase in handgrip strength of 1.58 kg (95% CI 0.17–3.0; *p* = 0.03) and concluded that TRT showed a beneficial effect on sarcopenic components, such as muscle mass and strength, as well as on physical performance in middle-aged and older adults [83]. Another recent systematic review also supports certain beneficial contributions of TRT to muscle condition and function [84].

However, there are limited data currently available regarding the direct effects of TRT on preventing sarcopenia, which is diagnosed based on muscle mass, strength, and physical functions, and the conclusions drawn from those data have been conflicting [50,69,72,80]. Two RCTs demonstrated that TRT contributed to an improvement in both muscle mass/strength and physical performance [69,72]. However, a previous study that included 209 hypogonadal men with mobility limitations reported that 6 months of TRT could increase muscle strength and stair-climbing power but could not improve walking speed [80]. Another study observed a significant increase in lean mass in the TRT group, whereas there were no differences in strength or physical performance between the control and TRT groups [50]. Furthermore, clinical studies targeted to Asian people, especially men, have been extremely limited. Further studies with large numbers of participants and various races are likely to reach a more definite conclusion regarding the effects of TRT on sarcopenia.

Studies have examined the dose-dependent effects of testosterone. A study that included healthy male adult participants randomly assigned to a weekly administration group (100 mg of testosterone enanthate weekly in an intramuscular injection) and a monthly group (alternating months of 100 mg of testosterone or placebo) for 5 months demonstrated that both groups had an increase in fiber diameter and peak power, with the weekly treatment being five-fold more effective than the monthly treatment [85]. In addition, a higher dose of testosterone can affect muscle mass and strength, not only for hypogonadal men but also eugonadal older men and healthy young men [86]. These data suggest that the anabolic effects of TRT are likely to be dose dependent to a certain extent. However, higher doses are associated with a high frequency of adverse effects and caution is required.

### 5.3. Other Systemic Effects of Testosterone Replacement Therapy

In general, TRT is rarely used to treat men solely for sarcopenia and is a widely accepted tool to improve various symptoms and clinical conditions occurred in hypogonadal men, including decreased libido and sexual desire, depression, muscle weakness, obesity, deterioration of insulin resistance, dyslipidemia, and osteoporosis [1,2,3,4]. Many RCTs and systematic reviews demonstrate that TRT can improve libido and sexual function, mood and energy, quality of life, anemia, bone density, cognitive function, body composition, in addition to muscle mass and strength [1,2,87,88,89,90]. In addition, some recent studies have supported the long-term use of TRT for 4 to 5 years to obtain beneficial effects for various metabolic parameters, body composition, and erectile function [91,92,93,94].

### 5.4. Adverse Effects and Risks of Testosterone Replacement Therapy

It is widely known that TRT is significantly associated with some adverse effects, such as erythrocytosis, gynecomastia, liver toxicity, testicular atrophy and infertility, acne, exacerbate sleep apnea, and potential growth of prostate cancer [1,87,88,95]. In particular, elderly men who are often candidates for TRT are originally at increased risk of prostate cancer. Therefore, men who receive TRT should undergo prostate-specific antigen screening regularly before and during treatment. On the other hand, there is no good evidence that testosterone administration can convert subclinical prostate cancer to clinically significant cancer and can increase risk of prostate cancer [87,95,96]. The association between TRT and cardiovascular risk is still controversial. Some recent meta-analyses did not demonstrate a significant association between TRT and any cardiovascular events [97,98].

## 6. Conclusions

Sarcopenia is an important pathophysiology factor of frailty in older adults and is diagnosed in older adults with decreased muscle strength, muscle mass, and walking speed, which can lead to a serious decrease in QOL. Testosterone directly interacts with the androgen receptor expressed in myonuclei and satellite cells and is then also indirectly associated with muscle metabolism through various cytokines and molecules. Significant correlations between testosterone and frailty in men have been confirmed by numerous cross-sectional studies. In addition, numerous RCTs have supported the beneficial effect of TRT on muscle mass and strength among men with low to normal testosterone levels. In the world’s aging society, TRT can be a tool for preventing the development of sarcopenia in older-adult men, although further RCTs are required.

## Figures and Tables

**Figure 1 jcm-11-06202-f001:**
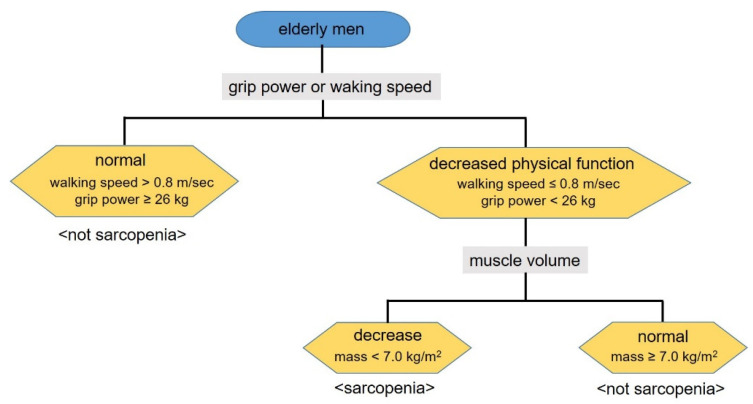
Algorithm for sarcopenia diagnosis (the Asian working group for sarcopenia).

**Figure 2 jcm-11-06202-f002:**
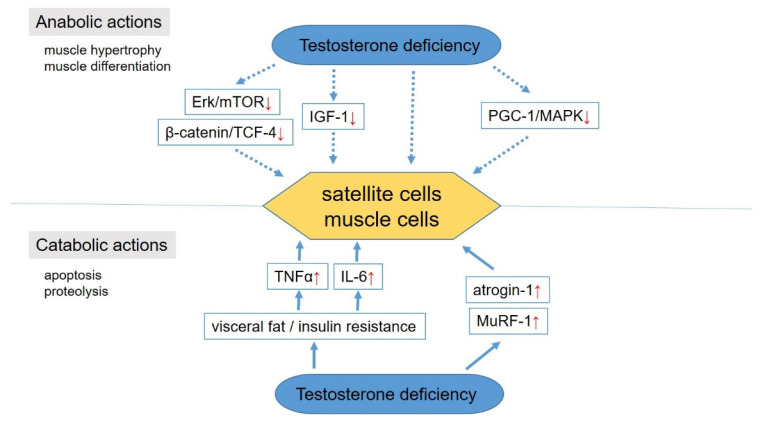
Molecular mechanisms in the development of sarcopenia. IGF-1, insulin-like growth factor–1; mTOR, mammalian target of rapamycin; TCF-4, T-cell factor-4; PGC-1α, peroxisome proliferator-activated receptor-gamma coactive tor 1 alpha; MAPK, p38 mitogen-activated protein kinases; TNF-α, tumor necrosis factor; IL-6, interleukin-6; MuRF-1, muscle atrophy-F-box (atrogin-1) and muscle RING-finger protein-1. An upward red arrow indicates “increase”, whereas a downward one indicates “decrease”.

**Table 1 jcm-11-06202-t001:** Classification of sarcopenia by causes.

Primary Sarcopenia	
Age-related	No clear causes other than aging
**Secondary Sarcopenia**	
Daily living-related	Bedridden, lack of exercise, ataxia, weight loss
Nutrition-related	Malabsorption, gastrointestinal disease, appetite loss, lack of energy, low protein intake
Disease-related	Severe organ failure, inflammatory diseases, malignancies, endocrine diseases

**Table 2 jcm-11-06202-t002:** Randomized control trials to investigate the effects of testosterone replacement therapy (TRT) on muscle (published since 2010).

Author	Year	Subjects	Number		TRT Regimens (Add-on Therapy)	Effects	Ref.
Kolind (Denmark)	2022	Hypogonadal men with opioid-treated chronic pain (TT < 12 nmol/L)	41	TRT 20 placebo 21	TU 1000 mg, intramuscular for 24 weeks	TRT did not improve muscle function (leg-press maximal voluntary contraction, leg extension power and handgrip strength).	[65]
Barnouin (USA)	2021	Hypogonadal men with obesity (TT < 10.4 nmol/L)	83	TRT 42 placebo 41	T gel daily for 6 months (diet + exercise)	TRT might attenuate the weight loss–induced reduction in muscle mass. There was no significant difference in muscle strength between the two groups.	[66]
Chasland (Australia)	2021	Men with obesity and low-normal serum TT (TT 6–14 nmol/L)	80	TRT 40 placebo 40	T gel 100 mg/day for 23 weeks (exercise)	TRT increased total, leg, and arm lean mass but did not affect aerobic capacity (Vo2peak) and muscle strength.	[67]
Glintborg (Denmark)	2020	Men with opioid-induced hypogonadism (TT < 12 nmol/L)	41	TRT 20 placebo 21	TU 1000 mg, intramuscular for 24 weeks	TRT increased lean body mass.	[68]
Gagliano-Juca (USA)	2018	Older men with mobility limitations (TT < 350 ng/dL or FT < 50 pg/mL)	99	TRT 46 placebo 53	T gel 100 mg/day for 6 months	TRT improved muscle strength and physical function (assessed by loaded stair-climbing power).	[69]
Storer (USA)	2017	Eugonadal and hypogonadal men (TT 100–400 ng/dL or FT < 50 pg/mL)	256	TRT 135 placebo 121	T gel 75 mg/day for 3 years	TRT strengthened chest-press strength and power, and leg-press power.	[70]
Ng Tang Fui (Australia)	2016	Hypogonadal men with obesity (TT < 12 nmol/L)	100	TRT 49 placebo 51	TU 1000 mg intramuscular for 56 weeks (diet)	TRT did not increase muscle volume but did attenuate the reduction in lean mass by diet compared with the controls.	[71]
Dias (USA)	2016	Hypogonadal men (TT < 350 ng/dL)	39	TRT 13 placebo 9 other 13	T gel 50 mg/day for 12 months	TRT improved knee strength and fast gait at 12 months compared with baseline.	[72]
Konaka (Japan)	2016	Hypogonadal men (FT < 10.8 pg/mL)	334	TRT 169 control 165	TE 250 mg/4 weeks for 52 weeks	TRT improved muscle volume and grip power.	[73]
Magnussen (Denmark)	2016	Hypogonadal men with DM (BioT < 7.3 nmol/L)	43	TRT 22 placebo 21	T gel 50 mg/day for 24 weeks	TRT increased lean body mass.	[74]
Sinclair (Australia)	2016	Hypogonadal men with cirrhosis (TT < 12 nmol/L or FT < 230 pmol/L)	101	TRT 50 placebo 51	TU 1000 mg/6–12 weeks intramuscular for 12 months	TRT increased total lean body and appendicular lean muscle mass.	[75]
Borst (USA)	2014	Hypogonadal men (TT ≤ 300 ng/dL)	60	TRT 31 placebo 29	TE 125 mg/weeks I intramuscular for 12 months (finasteride)	TRT increased upper and lower body muscle strength by 8–14% and fat-free mass by 4.04 kg.	[76]
Giamatti (Australia)	2014	Hypogonadal men with type 2 diabetes mellitus (TT ≤ 300 ng/dL or BioT ≤ 70 ng/dL)	88	TRT 45 placebo 43	TU 1000 mg/6–12 weeks intramuscular for 56 weeks	TRT increased lean body mass.	[77]
Stout (UK)	2012	Men with chronic heart failure (TT < 15 nmol/L)	28	TRT 15 placebo 13	Testosterone 100 mg/2 weeks intramuscular for 12 weeks (exercise)	TRT could not improve the shuttle walk test and hand grip strength compared with placebo.	[78]
Behre (Australia)	2012	LOH men (TT < 15 nmol/L or BioT < 6.68 nmol/L)	362	TRT 183 placebo 179	T gel 50–75 mg/day for 6 months	TRT increased lean body mass.	[79]
Travison (USA)	2011	Hypogonadal men with mobility limitation (TT 100–350 ng/dL or FT < 50 pg/mL)	209	TRT 106 placebo 103	T gel 100 mg/day for 6 months	TRT increased leg-press and chest-press strength and stair-climbing power but could not improve walking speed.	[80]
Atkinson (UK)	2010	Hypogonadal frail men (TT < 12 nmol/L)	30	TRT 16 placebo 14	T gel 50 mg/day For 6 months.	TRT helped preserve muscle thickness. There was no significant effect of treatment on fascicle length or pennation angle.	[81]
Kenny (USA)	2010	Hypogonadal frail men (TT < 350 ng/dL)	131	TRT 69 placebo 62	T gel 5 mg/day for 12–24 months	There was an increase in lean mass in the testosterone group but no differences in strength or physical performance.	[50]
Srinivas-Shankar (UK)	2010	Hypogonadal frail men (TT < 12 nmol/L or FT < 250 pmol/L)	274	TRT 138 placebo 136	T gel 50 mg/day for 6 months	Isometric knee extension peak torque was improved, and lean mass was increased in the TRT group.	[82]

TRT, testosterone replacement therapy; TT, total testosterone; FT, free testosterone; BioT, bioavailable testosterone; TU, testosterone undecanoate; TE, testosterone enanthate.

## Data Availability

Not applicable.

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
