# Peer review of "Relationship between Testosterone and Sarcopenia in Older-Adult Men: A Narrative Review"

_jcm, 2022, doi:10.3390/jcm11206202_

Round 1

Reviewer 1 Report

The manuscript entitled "Relationship between testosterone and sarcopenia in older 2

adult men; a narrative review" is interested and well written.

Limitations:

1- The authors should mention some more details about testesterone (chemical structure-function-mechanism).

2- Indication, Side effects, limitation of use of testosterone therapy .

Author Response

Thank you for reviewing our manuscript. In accordance with reviewer's comments, I have revised the manuscript. Revised points are indicated by red color.

  1. Details of testosterone, such as chemical functions, mechanism have been added. (Line 115-126)
  2. Indication, side effects, and limitation of use of testosterone therapy have been described in line 217-227, and line 292-313.

Reviewer 2 Report

unless a paragraph will present the potential side effects and safety concerns of long term Testosterone treatment in adults will be stated. Currently it is not included. 

Also, the authors should relate to cultural and ethnic diversity - are the findings they quate relating to a heterogenous enough population?

Author Response

Thank you for reviewing our manuscript. In accordance with reviewer's comments, I have revised the manuscript. Revised points are indicated by red color.

  1. Adverse effects and risks of testosterone therapy have been described in line 293-303.
  2. As the reviewer's comment, there may be some differences between races in effects of testosterone on sarcopenia/frailty! Certainly, clinical studies targeted to yellow race, especially Asian men, has been extremely limited. Therefore, I have added some comments. (Line 264-267).    

Reviewer 3 Report

I would avoid the term "Andropause" especially in the abstract as it invokes comparisons to a menopause. 

The paper is well written, but I would I include reference to the summary from the T trials which summarizes the other benefits of TTh in terms of sexual and cognitive function that usually go hand in hand with benefits in muscle as we are rarely treating men solely for sarcopenia.

It is a pity that the papers included stopped at 2020 and missed the T4DM study, Wittert et al with 1007 patients for 2 years, which would have virtually doubled the database with a 40% reduction in diabetes progression being highly important.

Author Response

Thank you for reviewing our manuscript. In accordance with reviewer's comments, I have revised the manuscript. Revised points are indicated by red color.

  1. A term 'andropause' is deleted in abstract.
  2. As reviewer's comments, TRT is rarely used to treat men solely for sarcopenia, and is a widely accepted tool to improve various symptoms and clinical conditions occurred in hypogonadal men. Therefore, I have added some descriptions regarding other systemic effects of TRT. (Line 280-290).